# Effects of a Sodium Phosphate Electrolyte Additive on Elevated Temperature Performance of Spinel Lithium Manganese Oxide Cathodes

**DOI:** 10.3390/ma14164670

**Published:** 2021-08-19

**Authors:** Minsang Jo, Seong-Hyo Park, Hochun Lee

**Affiliations:** 1Department of Energy Science and Engineering, Daegu Gyeongbuk Institute of Science and Technology (DGIST), Daegu 42988, Korea; alstkdwh@dgist.ac.kr (M.J.); serafpsh@dgist.ac.kr (S.-H.P.); 2Energy Science and Engineering Research Center, Daegu Gyeongbuk Institute of Science and Technology (DGIST), Daegu 42988, Korea

**Keywords:** lithium-ion battery, spinel lithium manganese oxide, manganese dissolution, sodium phosphate, electrolyte additive, thermal stability

## Abstract

LiMn_2_O_4_ (LMO) spinel cathode materials suffer from severe degradation at elevated temperatures because of Mn dissolution. In this research, monobasic sodium phosphate (NaH_2_PO_4_, P2) is examined as an electrolyte additive to mitigate Mn dissolution; thus, the thermal stability of the LMO cathode material is improved. The P2 additive considerably improves the cyclability and storage performances of LMO/graphite and LMO/LMO symmetric cells at 60 °C. We explain that P2 suppresses the hydrofluoric acid content in the electrolyte and forms a protective cathode electrolyte interphase layer, which mitigates the Mn dissolution behavior of the LMO cathode material. Considering its beneficial role, the P2 additive is a useful additive for spinel LMO cathodes that suffer from severe Mn dissolution.

## 1. Introduction

Although lithium-ion batteries (LIBs) have been successfully commercialized, to fully adopt them in mobile devices, electric vehicles (EVs), hybrid electric vehicles (HEVs), and grid-level energy storage applications, a remarkable improvement in the long-term reliability and safety under abuse conditions is urgently required [1,2,3]. Several types of cathode materials have been employed for LIBs [4,5,6], such as the layered-type, LiCoO_2_ (LCO) and LiNi_x_Co_y_Mn_z_O_2_ (NCM); spinel-type, LiMn_2_O_4_ (LMO) and LiMn_1.5_Ni_0.5_O_4_ (LMNO); and olivine-type, LiFePO_4_ (LFP). Among the cathode materials, LMO is a promising cathode material because of its superior electrochemical performance, low cost, high safety, and environmental inertness [7,8,9].

However, the commercial application of LMO cathodes is hindered by rapid capacity fading and poor cyclability at elevated temperatures [7]. Electrolytes containing lithium hexafluorophosphate (LiPF_6_) are unavoidably contaminated by a detrimental byproduct: hydrofluoric acid (HF), which causes fatal Mn dissolution from the LMO cathode and undesired side reactions on the anode [10]. It is generally known that Mn dissolution is associated with a disproportionation that liberates soluble Mn^2+^ into the electrolyte: 2Mn^3+^ (electrode) → Mn^4+^ (electrode) + Mn^2+^ (electrolyte) [11]. The dissolved Mn^2+^ can be deposited on the anode surface, which exacerbates the degradation of the anode/electrolyte interface. This has been reported to be a common failure mechanism of the LMO-based LIBs at elevated temperatures [10,12,13].

To improve the thermal stability of LMO-based cells, cathode modification methods, such as doping with foreign ions [14,15,16,17] and surface coating with oxides [18,19,20] and polymers [21,22], have been reported. However, these methods require complex fabrication processes and result in inhomogeneous surfaces and cost issues. In contrast, the application of functional electrolyte additives is a more economical and effective method. Various functional electrolyte additives have been reported to enhance the high-temperature stability of LMO-based cells, such as fluoroethylene carbonate, which forms a stable and thin solid electrolyte interphase (SEI) layer on the graphite anode [23], hexamethyldisilazane and heptamethyldisilazane, which scavenge H_2_O and HF in the electrolyte [7,24], and lithium salts, including lithium difluoroborate and lithium bis(oxalato)borate, which form a stable SEI layer on the LMO cathode [25]. These electrolyte additives have been reported to effectively prevent Mn dissolution from the LMO cathode and considerably maintain the spinel structure. However, the cell performance improvement with a single electrolyte additive is limited; research is needed into multifunctional electrolyte additives, which scavenge HF in the electrolyte as well as form a stable CEI layer, for LMO-based cells.

Recently, our group reported that monobasic sodium phosphate (NaH_2_PO_4_, P2) significantly enhances the thermal stability and cycle performance of LiNi_0.8_Co_0.1_Mn_0.1_O_2_ (NCM811)/graphite cells at 60 °C [26]. We revealed that P2 forms a protective CEI layer that prevents NiO rock salt structure formation on the NCM811 cathode as well as electrolyte decomposition, which is a root cause of the degradation of the Ni-rich NCM cathodes. Furthermore, it scavenges HF in the electrolyte, and thus, the degradation of the SEI layer on the anode is suppressed. In this context, P2 is expected to also enhance the performance of the LMO cathode, which is adversely affected by HF in the electrolyte and exhibits poor cyclability at elevated temperatures. Inspired by our earlier work, for the first time, P2 is investigated as an electrolyte additive for the LMO cathodes in this study.

## 2. Materials and Methods

### 2.1. Materials and Electrode Preparation

Battery-grade ethylene carbonate (EC), ethyl methyl carbonate (EMC), and lithium hexafluorophosphate (LiPF_6_) were obtained from Panax Etec (Nonsan, Korea). P2 (>99.0%, Sigma-Aldrich) was used as received. A mixture of EC and EMC (3/7, *v*/*v*) containing 1.0 M LiPF_6_ was used as the base electrolyte, and the base electrolyte saturated with P2 (P2 content = 0.4 ± 0.1 wt%) was used as the P2 electrolyte. All the electrolytes were prepared in an Ar-filled glove box (≤5 ppm of H_2_O and O_2_). The cathode electrode was fabricated on Al foil using LiMn_2_O_4_ (90 wt%, POSCO ESM, hereafter LMO), conductive carbon (5 wt%), and polyvinylidene difluoride (PVDF) binder (5 wt%). The graphite electrode was fabricated on a Cu foil with artificial graphite (95 wt%), conductive carbon (3 wt%), and PVDF binder (2 wt%). The areal capacity of the LMO cathode was 1.3 mAh cm^−2^, and that of the graphite anode was 1.5 mAh cm^−2^.

### 2.2. Electrochemical Analysis

A three-electrode configuration was used for linear sweep voltammetry (LSV) to examine the oxidation behavior of the electrolytes. Pt disk (area: 0.02 cm^2^), Pt wire, and Li foil were used as the working, counter, and reference electrodes, respectively. The scan rate was 1 mV s^−1^. LSV experiments were conducted in the glove box using a potentiostat (VSP-300, BioLogic, Seyssinet-Pariset, France). The differential capacity versus potential curves (dQ/dV vs. V) were observed during the first lithiation process of graphite/Li coin cells at 0.1 C current to examine the reduction behavior of the base and P2 electrolytes.

For the battery cycling tests, LMO/graphite, LMO/LMO, and graphite/graphite cells were assembled in 2032-type coin cells using the prepared electrodes with a polyethylene (PE) separator (Tonen, 20 µm thickness, Tokyo, Japan) in the glovebox. The cycle performance tests for LMO/graphite cells were conducted at 0.5 C constant current–constant voltage (CC-CV) charging and 0.5 C CC discharging over 3.0–4.3 V at 60 °C. LMO/LMO and graphite/graphite cells were charged and discharged at 60 °C at 0.5 C CC-CV over ±1.3 V and ±0.8 V, respectively. For the storage tests at elevated temperature, the open-circuit voltage (OCV) of the fully charged LMO/graphite cells was monitored while stored at 60 °C. To examine the discharge rate capability, the LMO/graphite cells were CC-CV charged at 0.2 C and discharged at various current densities from 0.5 C to 5.0 C at 25 °C. LMO/LMO and graphite/graphite symmetric cells were fabricated using LMO or graphite electrodes with a state-of-charge (SOC) of 50, collected from LMO/Li or graphite/Li cells that had been cycled three times over 3.0–4.3 V (LMO/Li cells) or 1.5–0.005 V (graphite/Li cells), respectively. A battery tester (Toscat-3000, Toyo System, Fukushima, Japan) equipped with temperature chambers was used for the battery charge/discharge tests. Electrochemical impedance spectroscopy (EIS) analysis was performed for the LMO/graphite cells using the potentiostat in the frequency range between 300 kHz and 10 mHz at an amplitude of 5 mV at 25 °C.

### 2.3. Material Characterization

Field-emission transmission electron microscopy (FE-TEM, HF3300, Hitachi, Tokyo, Japan) and field-emission scanning electron microscopy (FE-SEM, S-4800, Hitachi) were used to examine the surface and cross-sectional morphologies of the LMO electrodes before and after cycling. X-ray photoelectron spectroscopy (XPS, ESCALAB 250Xi, Thermo Fisher Scientific Inc., Waltham, MA, USA) with a monochromatic Al K_α_ source was used to examine the surface compositions of the LMO and graphite electrodes. The LMO and graphite electrodes were collected from the LMO/graphite cells after cycled 100 times (TEM and SEM) or three times (XPS), washed with dimethyl carbonate, followed by being dried in the glove box.

To investigate the Mn dissolution behavior, the LMO electrode (diameter: 14 mm) with the electrolyte (4 mL) in polytetrafluorethylene (PTFE) bottles was preserved at 60 °C for 48 h. The electrolyte was diluted with distilled water, and the dissolved Mn concentration was analyzed by atomic absorption spectroscopy (AAS, AA-7000, Shimadzu Corporation, Tokyo, Japan).

## 3. Results and Discussion

Table 1 summarizes the preliminary results on the effects of some selected electrolyte additives on the Mn dissolution behavior. The pristine LMO cathodes were preserved in 1.0 M LiPF_6_-EC/EMC (3/7, *v*/*v*) electrolyte containing 2 wt% of additives at 60 °C for 48 h, and the quantity of dissolved Mn ion was compared. In this study, P2 was chosen as an electrolyte additive for the LMO cathode because the electrolyte containing P2 exhibited the lowest Mn concentration after the storage test, which indicates that P2 most effectively suppressed Mn dissolution from the LMO cathode. As mentioned in the introduction section, Mn dissolution can result from HF in the electrolyte, and thus, Mn dissolution can be suppressed using an additive that can scavenge HF in the electrolyte. Our previous study revealed that P2 can scavenge HF in the electrolyte, and the results are summarized in Appendix A [26] (the latter designated as Appendix A). As shown in Appendix A, HF reduction peak was observed between 2.4 and 3.0 V (vs. Li/Li^+^) during the first cathodic voltammetric scan, and the intensity of the peak current is in proportion with the HF concentration. The peak currents shown in Appendix A are summarized in Appendix A. P2 significantly decreased the peak current when the HF content was higher than 200 ppm. This suggests that P2 can scavenge HF, and the process accelerates as HF content increases.

Our previous study revealed that a cathode electrolyte interphase (CEI) layer can be formed by the oxidation decomposition of P2, and it suppresses further oxidation of the electrolyte, which was confirmed by LSV on a Pt disk electrode [26]. As shown in Appendix A, the oxidation current above ca. 4.0 V with the P2 electrolyte was a bit higher than that of the base electrolyte at the first anodic scan. Meanwhile, almost similar LSV curves were observed for the both electrolytes in the second scan (inset of Appendix A). This behavior implies that a CEI layer was formed by the oxidation of P2, and mitigated the further oxidation of the electrolyte. Conversely, our previous study confirmed that P2 has a negligible effect on the SEI layer formation on the graphite anode, as depicted in Appendix A [26].

To examine the compositional difference of the CEI layers, XPS measurements were conducted for the LMO cathodes after three cycles in the base and P2 electrolytes. Figure 1a,b displays the XPS spectra of the LMO cathodes with the base and P2 electrolytes. As shown in the Na 1s spectra (Figure 1a), no distinct peak was observed for the LMO cathode with the base electrolyte. Meanwhile, a new peak was observed at 1072.6 eV for the cathode with the P2 electrolyte. Conversely, as shown in Figure 1b, a peak at 136.2 eV was observed in the P 2p spectra for the LMO cathode with both electrolytes. This peak was assigned to the decomposition products of LiPF_6_ such as Li_x_PF_y_ [27,28]. In addition, an additional peak at 134.0 eV was detected in the cathode with the P2 electrolyte, while this peak was not detected in the cathode with the base electrolyte. These new signals in the Na 1s and P 2p spectra can be assigned to the decomposition product of P2 (e.g., P-O moiety) [29,30], indicating that the P2-derived CEI layer was formed on the LMO cathode surface by the oxidation of the electrolyte. Contrarily, no distinct peak was observed in the Na 1s spectra (Figure 1c), and two peaks were observed in the P 2p spectra (Li_x_PO_y_F_z_ at 133.7 eV and Li_x_PF_y_ at 136.2 eV, Figure 1d) for both the graphite anodes cycled in the base and P2 electrolytes. In other words, the presence of P2 hardly affects the Na 1s and P 2p spectra of the graphite anode. This result suggests that the impact of the P2 additive on the composition of the SEI layer on the graphite anode was negligible.

In addition, the F 1s spectra displayed in Appendix A show the LiF formed by the reaction of HF with Li^+^ ions or the LMO cathode materials. It has been reported that HF in the electrolyte can form LiF by two mechanisms: the formation of LiF by consuming Li ions (HF + Li^+^ + e^−^ → LiF + 1/2H_2_) and the dissolution of Mn ions from the LMO cathode as well as the formation of LiF (LiMn_2_O_4_ + xHF → Li_1−x_Mn_2−y_O_4−x/2_ + xLiF + yMn^2+^ + x/2H_2_O) [27,31,32]. The LiF peak intensity of the LMO cathode with the P2 electrolyte distinctly decreased compared to that with the base electrolyte (Appendix A). This suggests that LiF formation was effectively mitigated by P2 which eliminates HF from the electrolyte. As shown in Appendix A, the F 1s spectra of the graphite anode also showed reduced LiF peak intensity, which can be attributed to the suppression of SEI layer degradation due to HF (e.g., 2HF + Li_2_CO_3_ → 2LiF + H_2_O + CO_2_) [33,34].

Based on the ability of P2 to scavenge HF impurities in the electrolyte and form a CEI layer on the LMO cathode, the effect of P2 on the cycling stability of the LMO cathode at elevated temperatures was examined. Figure 2a shows the cyclability of the LMO/graphite cells at 60 °C. The LMO/graphite cell with the P2 electrolyte exhibited significantly improved capacity retention compared to the cell with the base electrolyte (44.1% vs. 15.5% after 200 cycles). In addition, Figure 2b compares the accumulated irreversible specific capacity, which is defined as the sum of the irreversible specific capacity that occurred over the cycle (Σ(Q_i_^c^ − Q_i_^d^), where Q_i_^c^ and Q_i_^d^ are the charge and discharge specific capacities at the *i*th cycle). It has been reported that the number of parasitic side reactions that occur during the charge/discharge process can be quantified by measuring the accumulated irreversible specific capacity [35,36]. The cell with the P2 electrolyte exhibited lower accumulated irreversible specific capacity than the cell with the base electrolyte (73.2 mAh g^−1^ vs. 144.1 mAh g^−1^ after 200 cycles). This implies that P2 can suppress the occurrence of the parasitic side reactions upon cycling by forming a protective P2-derived CEI layer, as well as using its HF scavenging ability.

Figure 3 shows the TEM and SEM images of the LMO cathodes before and after 100 cycles at 60 °C. No surface layer was observed for the uncycled, pristine LMO cathode (Figure 3a); however, the LMO cathodes subjected to being cycled 100 times at 60 °C with the base and P2 electrolytes showed a tenth-of-nanometer-thick CEI layer formed on their surface (highlighted by red dotted lines in Figure 3b,c). However, the morphologies of the CEI layers formed in the base and P2 electrolytes were not significantly different. Furthermore, the CEI layer composition on the LMO cathodes obtained from TEM/EDX analysis is summarized in Appendix A. The Na content of the CEI layer was 1.29 wt% for the LMO cycled with the P2 electrolyte, while it was not detected on that with the base electrolyte. This suggests that the P2-derived CEI layer was formed on the LMO cycled with the P2 electrolyte. Figure 3d–f shows the cross-sectional SEM images of the LMO electrodes before cycling (pristine) and those collected from the LMO/graphite cells after being cycled 100 times at 60 °C with the base and P2 electrolytes. Compared to the pristine case (Figure 3d), the LMO electrode cycled with the base electrolyte (Figure 3e) exhibited extensive crack formation. By contrast, the electrode cycled with P2 electrolyte considerably preserved the electrode structure (Figure 3f).

The self-discharge behavior during storage at 60 °C was compared for the fully charged LMO/graphite cells with and without P2 to cross-check the enhanced thermal stability enabled by the P2 additive. It has been reported that the dissolution of Mn ions from the cathode and their subsequent deposition on the anode can affect the OCV during high-temperature storage [37,38]. As shown in Figure 4a, the P2 electrolyte significantly delayed the OCV drop of the cell compared to that with the base electrolyte (the cell with the base and P2 electrolyte took 181 h and 336 h, respectively, to reach 4.0 *V*). Furthermore, the presence of Mn deposits on the graphite anodes was examined using the XPS measurements of the graphite anodes retrieved from the LMO/graphite cells after the high-temperature storage test (Figure 4b). The graphite anode collected from an LMO/graphite cell with the base electrolyte presented a distinct signal at 642.7 eV in the Mn 2p spectra. In contrast, the Mn 2p peak was absent in the graphite anode with the P2 electrolyte. These results suggest that Mn dissolution and following deposition on the anode were suppressed in the cell with the P2 electrolyte during the high-temperature storage test.

To distinguish the beneficial role of the P2 additive on the LMO cathode and graphite anode, the cyclability of the LMO/LMO and graphite/graphite symmetric cells at 60 °C was examined. The capacity retention of the LMO/LMO symmetric cells at a 0.5 C current during 100 cycles drastically improved from 57.2% to 85.2% when using the P2 electrolyte (Figure 5a). As shown in Figure 5b, unlike the LMO/LMO symmetric cells, the graphite/graphite symmetric cells showed slightly enhanced capacity retention with the P2 electrolyte compared to that of the base electrolyte (52.9% vs. 48.9% at 100 cycles). This slightly enhanced cyclability of the graphite/graphite symmetric cells was attributed to the reduced amount of HF in the electrolyte that suppressed the degradation of the SEI layer, although the composition of the SEI layer on the graphite anode was insignificantly changed by the addition of P2, as confirmed in the previous section. Hence, the enhanced cycle performance of the LMO/graphite cell with the P2 electrolyte (Figure 2) can be mainly attributed to the beneficial effects of P2 on the LMO cathode rather than the graphite anode.

Moreover, to assess the rate capability of LMO/graphite cells with the base and P2 electrolytes, the specific capacities were measured at different discharge rates up to 5 C at 25 °C (Appendix A). The cell with the P2 electrolyte showed a slightly enhanced rate capability compared to the base electrolyte. Appendix A shows the EIS spectra of the LMO/graphite cells with the base and P2 electrolytes at an SOC of 100. A distorted semicircle, which consisted of several semicircles with similar time constants, was observed for the base electrolyte [39]. Meanwhile, three semicircles appeared for the cell with the P2 electrolyte, which was attributed to the resistance of the CEI and SEI films, the electronic properties of the material, and the charge transfer resistance, respectively [40,41]. The total cell resistance was significantly reduced by the P2 additive, and this implies that the improved rate capability of the LMO/graphite cell, as shown in Appendix A**,** is attributed to the P2-derived CEI layer with relatively low resistance.

## 4. Conclusions

This study demonstrated that the P2 electrolyte additive improved the cycle performance and thermal stability of the LMO/graphite cell at 60 °C. P2 eliminated HF from the electrolyte, modified the CEI layer on the LMO cathode, and suppressed Mn dissolution from the cathode without significantly affecting the SEI layer on the graphite anode. The cycle performance at 60 °C and storage performance of the LMO/graphite cell were dramatically improved by P2 addition. With P2 addition, the capacity retention of the LMO/graphite cell was improved from 15.5% to 44.1% after 200 cycles at 0.5 C current and 60 °C. The OCV drop of the LMO/graphite cell was significantly delayed during storage at 60 °C with the P2 electrolyte compared to the cell without P2. Considering the formidable advantages, P2 is a promising electrolyte additive for LMO cathodes that suffer from inferior cycling and storage performance at elevated temperatures. Furthermore, further research is underway to employ P2 as an electrolyte additive for the LNMO cathode, which suffers from Ni and Mn dissolution and severe capacity fading at elevated temperatures.

## Figures and Tables

**Figure 1 materials-14-04670-f001:**
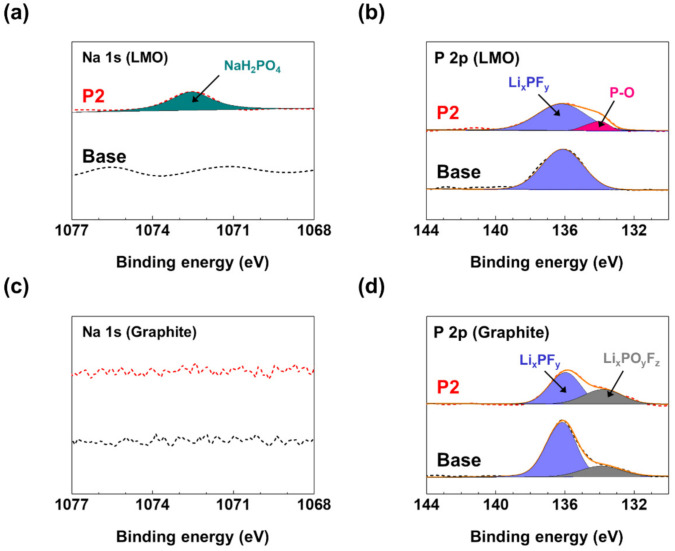
(**a**) Na 1s and (**b**) P 2p XPS spectra of the LMO cathodes with and without P2 additive. (**c**) Na 1s and (**d**) P 2p XPS spectra of the graphite anodes with and without P2 additive. All samples were collected from the LMO/graphite cells after three cycles. Dotted lines denote the experimental spectra and solid lines denote the best-fitted results.

**Figure 2 materials-14-04670-f002:**
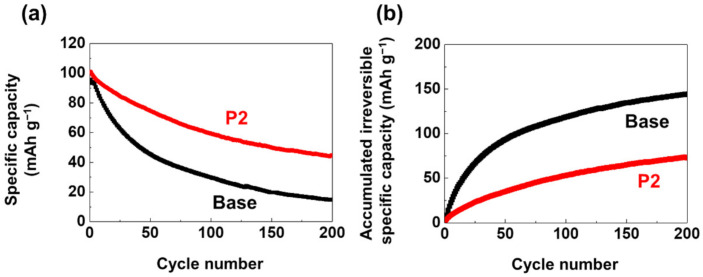
(**a**) The cycle performance and (**b**) accumulated irreversible specific capacity of LMO/graphite cells at 60 °C. The cells were charged and discharged at a 0.5 C current.

**Figure 3 materials-14-04670-f003:**
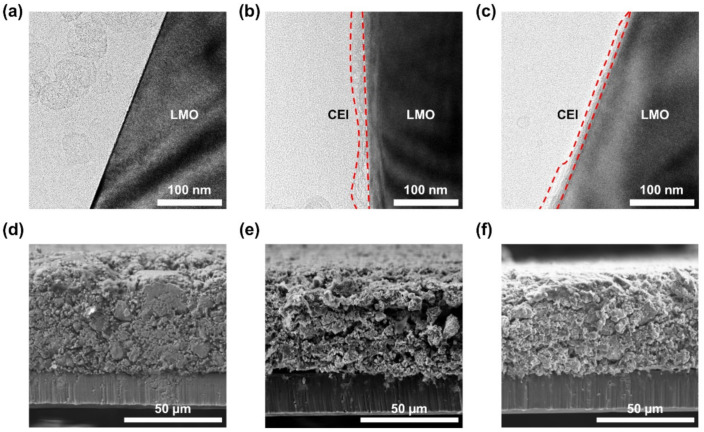
TEM images of LMO cathodes (**a**) before cycling and after cycling 100 times with (**b**) the base and (**c**) P2 electrolytes. Cross-sectional SEM images of LMO cathodes (**d**) before cycling and after cycling 100 times with (**e**) the base and (**f**) P2 electrolytes.

**Figure 4 materials-14-04670-f004:**
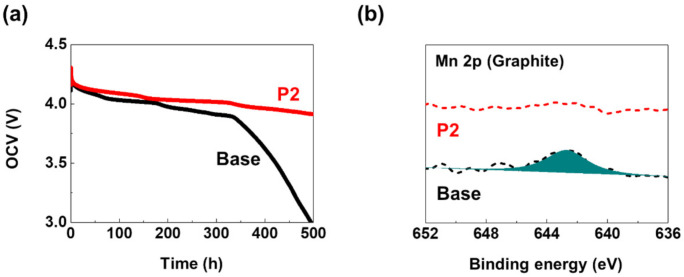
(**a**) OCV variation with time of LMO/graphite cells with the base and P2 electrolytes stored at 60 °C. (**b**) XPS Mn 2p spectra for the graphite electrodes after stored in the base and P2 electrolytes for 48 h at 60 °C. Dotted lines denote the experimental spectra and solid lines denote the best-fitted results.

**Figure 5 materials-14-04670-f005:**
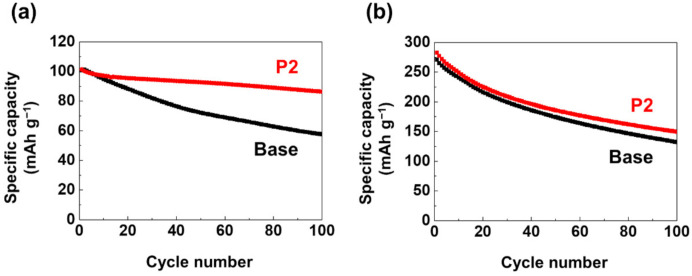
Cycle performance of (**a**) LMO/LMO and (**b**) graphite/graphite symmetric cells at 60 °C. The cells were charged and discharged at 0.5 C over ±1.3 V for LMO/LMO and ±0.8 V for graphite/graphite symmetric cells.

**Table 1 materials-14-04670-t001:** Comparison of dissolved Mn concentration with various electrolyte additives after the storage test.

Name ^1^	Structure	Mn Concentration (ppm)
base	-	0.345
VC	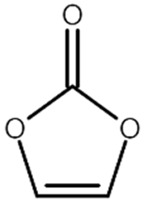	0.327
FEC	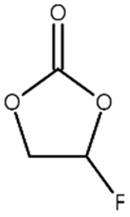	0.332
AB	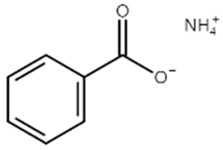	0.319
SN	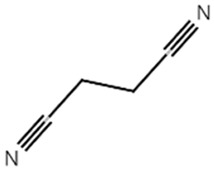	0.421
P1	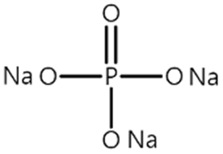	0.447
P2	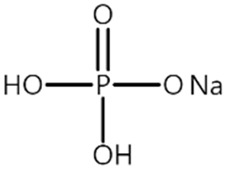	0.139
P3	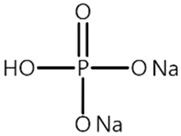	0.301
SS1	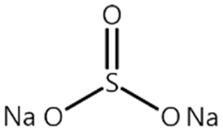	0.719

^1^ base: 1.0 M LiPF_6_-EC/EMC (3/7, *v*/*v*), VC: vinylene carbonate, FEC: fluoroethylene carbonate, AB: ammonium benzoate, SN: succinonitrile, P1: tribasic sodium phosphate, P2: monobasic sodium phosphate, P3: dibasic sodium phosphate, SS1: sodium sulfite.

## Data Availability

The data presented in this study are available on request from the corresponding author.

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
