# Peer review of "Effects of a Sodium Phosphate Electrolyte Additive on Elevated Temperature Performance of Spinel Lithium Manganese Oxide Cathodes"

_materials, 2021, doi:10.3390/ma14164670_

Round 1

Reviewer 1 Report

This work investigated the additive (NaH2PO4)) effect on the high-temperature performance of LiMn2O4. The results proved that NaH2PO4 is beneficial for suppressing Mn dissolution by forming a good CEI layer. It is potentially suitable for publishing in Materials after addressing following issues.

(1) Will Na+ intercalate into LiMn2O4 if using NaH2PO4 as the additive? X.-J. Nie et al. Electrochimica Acta 320 (2019) 1346263.

(2) What is the chemical composition of CEI layer? A TEM-EDX should be provided.

(3) the charge/discharge curves of LMO/LMO and Graphite/Graphite symmetric cells should be provided with proper discussion. Why choosing 0.5V for cycling?

(4) It is claimed that Mn dissolution was suppressed by NaH2PO4 additives. To prove this, atomic absorption spectroscopy of electrolytes after different cycles (1, 5, 10, 50, 100) should be provided.

Reviewer 2 Report

This paper is well arranged and good writing. It can be accepted now.

One more, the introduction should clerify the research motivation clearly. Please modify it.

Reviewer 3 Report

Line 24: …them in electric vehicles (EVs) and grid-level energy storage applications, - and what about the application of LIBs in the hybrid and classic vehicles and individual devices?

Line 30: it would be useful to mention the possibilities and problems of recycling in relation to particular types of batteries mentioned

Lines 31-35 – references needed for each of the statements

it would be worth taking a stance on the influence of Sodium Phosphate Electrolyte Additive on the formation of lithium dendrites in LMO cells - whether it occurs, is absent or not tested

Line 75: The specific capacity of the LMO cathode and graphite anode was 1.3 mAh cm−2 and 1.5 mAh cm−2, respectively. – It should be: The areal capacity [Li, Jie & Leu, Ming & Panat, Rahul & Park, Jonghyun. (2017). A hybrid three-dimensionally structured electrode for lithium-ion batteries via 3D printing. Materials & Design. 119. 417-424. 10.1016/j.matdes.2017.01.088] of the LMO cathode and graphite anode was 1.3 mAh cm−2 and 1.5 mAh cm−2, respectively.

Line 132: in Figure S1 [26]. – I suggest changing it into: in Figure S1 [26] (the latter designated as the ‘[1]’ in the Supplementary Materials).

Table 1: I suggest using SS1 instead S1, as the designation S1 for Figure S1 was used in the supplementary materials

Due to the use of the expression C as the capacity of a cell/battery, I suggest changing the temperature from the Celsius scale to the Kelvin scale in all cases.

Figure 1, Figures S3a, S3b - a description of the units on the vertical axis is needed, separately for individual graphs - in this case, I advise you not to accumulate two waveforms in one figure

Lines 196-198: accumulated irreversible capacity – rather: accumulated irreversible specific capacity

Figure 2: in the description of units on the vertical axis, the ‘specific capacity’ should be used instead of the ‘capacity’

it would be useful to compare the obtained results with those for a similar type of cells described in the literature, grafting in relation to capacity and cyclability

Conclusions – it is rather Summary. It would be useful to indicate here the directions of planned further research

Round 2

Reviewer 1 Report

This manuscript has been improved during the revision process. Now it can be published as it is.